# Establishment of an Efficient *Agrobacterium*-Mediated Genetic Transformation System to Enhance the Tolerance of the Paraquat Stress in Engineering Goosegrass (*Eleusine Indica* L.)

**DOI:** 10.3390/ijms24076629

**Published:** 2023-04-01

**Authors:** Qiyu Luo, Shu Chen, Hai Nian, Qibing Ma, Yuyao Ding, Qinwen Hao, Jiping Wei, Jinesh D. Patel, Joseph Scott McElroy, Yaoguang Liu, Yong Chen

**Affiliations:** 1College of Agriculture, South China Agricultural University, Guangzhou 510642, China; 2College of Life Sciences, South China Agricultural University, Guangzhou 510642, China; 3Department of Crop, Soil, and Environmental Sciences, Auburn University, Auburn, AL 36830, USA; 4College of Forestry and Landscape Architecture, South China Agricultural University, Guangzhou 510642, China

**Keywords:** genetic transformation, *Agrobacterium*, herbicide stress, paraquat-resistance, goosegrass

## Abstract

*Eleusine indica* (goosegrass) is a problematic weed worldwide known for its multi-herbicide tolerance/resistance biotype. However, a genetic transformation method in goosegrass has not been successfully established, making a bottleneck for functional genomics studies in this species. Here, we report a successful *Agrobacterium*-mediated transformation method for goosegrass. Firstly, we optimized conditions for breaking seed dormancy and increasing seed germination rate. A higher callus induction rate from germinated seeds was obtained in N6 than in MS or B5 medium. Then the optimal transformation efficiency of the *gus* reporter gene was obtained by infection with *Agrobacterium tumefaciens* culture of OD_600_ = 0.5 for 30 min, followed by 3 days of co-cultivation with 300 μmol/L acetosyringone. Concentrations of 20 mg L^−1^ kanamycin and 100 mg L^−1^ timentin were used to select the transformed calli. The optimal rate of regeneration of the calli was generated by using 0.50 mg L^−1^ 6-BA and 0.50 mg L^−1^ KT in the culture medium. Then, using this transformation method, we overexpressed the paraquat-resistant *EiKCS* gene into a paraquat-susceptible goosegrass biotype MZ04 and confirmed the stable inheritance of paraquat-resistance in the transgenic goosegrass lines. This approach may provide a potential mechanism for the evolution of paraquat-resistant goosegrass and a promising gene for the manipulation of paraquat-resistance plants. This study is novel and valuable in future research using similar methods for herbicide resistance.

## 1. Introduction

Grasses were used for food, feed, and beverages by humans 2 million years ago. Some of the grasses have been domesticated by humans as crops for agriculture [1]. Moreover, some of the most devastating agricultural weeds in crop fields also are grasses [2]. Recently, reviews have been made on expounding the grass research and addressing the value of grass diversity in the future human life [3,4,5,6]. However, the global herbicide resistant weeds found in 267 plant species (154 dicots and 113 monocots) have become a serious problem for agricultural production [7].

Goosegrass (*Eleusine indica* L. Gaertn) is one of the top 10 herbicide-resistant weed species distributed throughout the world [8,9]. Although multiple resistances to paraquat, glufosinate, and glyphosate in goosegrass have been reported, the underlying molecular mechanism of herbicide stress responses remains unclear [10,11]. One of the crucial reasons is that the genetic transformation system in goosegrass has not been established. The genetic transformation system in goosegrass can provide a deeper understanding of how resistant-gene expression responds to herbicide stress through its metabolism to help select functioning genes in the application of other engineering crops with multiple herbicide tolerance.

The genetic transformation of goosegrass followed the principles that rely on a combination of several factors including explants, genetic backgrounds, regulators of the growth of tissue culture, and regulators of *Agrobacterium*-mediated infection and regeneration. So far, only a few methods for callus induction, cell suspension culture, and plant regeneration of goosegrass from seeds, mature embryos, immature inflorescences, immature embryos, and young seedlings have been reported in the last decades [12,13,14,15,16]. In terms of callus types, they were defined and classified from type I to type III in switchgrass cultivars [17]. Moreover, we found that these types were consistent with the callus types of goosegrass.

Paraquat (1,1′-dimethyl-4,4′-bipyridinium dichloride) is one of the most widely used broad-spectrum, quick-acting, and nonselective herbicide [18]. Although a mutation of DTX6 was confirmed in *Arabidopsis* to enhance paraquat resistance, the genetic loci of weeds for paraquat resistance still lack the identification [19,20]. However, it was found that the structure of paraquat is similar to that of polyamines, which could share a common uptake system of endogenous substrates due to their structural similarity [21,22,23,24]. Recently, three genes on polyamine uptake transportation were applied to enhance the tolerance of paraquat-resistance in rice [25]. Therefore, it is worth exploring the mechanism of polyamine genes regulating paraquat resistance.

Our preliminary studies on paraquat-resistance goosegrass determined the paraquat dose-response and screened out four putative genes associated with resistant phenotypes and physiological indexes under paraquat stresses [26,27]. Moreover, we found that the treatments of exogenous spermidine appeared to protect the paraquat-susceptible goosegrass in response to salt stress [28]. We further overexpressed one of the putative genes *EiKCS* (encoding a β-ketoacyl-CoA synthase) in transgenic rice to enhance a paraquat-resistance and find its function on polyamine biosynthesis [29]. However, we still did not prove that *EiKCS* was the paraquat-resistance exclusively responsible. Therefore, it is necessary to confirm that the results of paraquat resistance in transgenic goosegrass are parallel to the observations in transgenic rice. In addition, more insight is necessary concerning how the *EiKCS* affects paraquat resistance via its polyamine metabolism in this weed species.

In this study, we established an efficient *Agrobacterium*-mediated transformation method for goosegrass, using calli induced from mature seeds of goosegrass. With this method, we successfully obtained a transgenic goosegrass by inducing it from the calli of paraquat-susceptible goosegrass. The transgenic goosegrass was monitored for the phenotypes of paraquat resistance and the changes in overexpression of the *EiKCS* gene and its protein EiKCS were confirmed. This study also presents our stepwise detailed protocol for the goosegrass transformation process. Thus, the ability to transform an important species may be useful in probing the physiology and metabolism of this weed. Likewise, this methodical approach is useful in that it may be applied to other difficult weed species.

## 2. Results and Discussion

### 2.1. Effects of the Explant and Genetic Backgrounds in Goosegrass Callus Induction

To evaluate the callus induction efficiency of explant types, different tissues of goosegrass were used as explants for callus induction. Among the used tissues of seeds, stems, leaves, and roots, the seeds showed the highest efficiency of callus induction (Figure 1A). Seeds of three paraquat-sensitive biotypes (MZ04, PY07, and QY04) showed the average callus induction rates of 78.86%, 66.67%, and 53.50%, respectively (Figure 1B). In addition, the frequency of induction rates of the MZ04 seeds was distributed above 75% (Figure 1C). Besides, the seeds were cultured in callus induction media of N6, MS, and B5, and those in the N6 medium showed the highest induction rate of 60.09% (Figure 1D). Then, we classified the induced calli into type-I to III in the N6 medium (Figure 1E). The types-I calli mainly had yellow and compact texture that could develop into embryogenic calli (EC). However, the type-II; and type-III calli were mainly milky-white friable and nodular textures that mostly became non-embryogenic calli (NEC). Moreover, the EC could rapidly enter the proliferation and differentiation processes after one or two subcultures (Figure 1F).

### 2.2. Effects of the Developmental Regulators on Goosegrass Calli Induction from Seeds

The effects of the developmental regulators showed that the treatment of 30% NaOH with 5–20 min greatly increased the germination rates (up to 26.25%) (Figure 2A). However, the treatment of −20 °C could also improve the germination rates up to 49% (Figure 2B). The seeds were also treated with 75% ethanol for 1–5 min followed by 0.10% HgCl_2_ for 15 min. Although the results did not show that one minute was significantly different than 2 or 5 min, the treatment with 75% ethanol solution for 1 min (and then with 0.10% HgCl_2_ for 15 min) had the highest germination rate (32%) without seed contamination (Figure 2C). To further reduce any negative influence of the induction rate from HgCl_2_ sterilization, 0.05%, 0.10%, and 0.20% concentrations of HgCl_2_ solution were used to sterilize the seeds for 10, 15, and 20 min. The callus induction rates of the seeds under sterilization with 0.05% HgCl_2_ trended higher than those with 0.10% and 0.20% HgCl_2_ (Figure 2D). The developmental regulators may improve callus induction in plants. As one of the critical regulators on callus induction, the use of 0.50 mg L^−1^ 2,4-D (2,4-dichlorophenoxyacetic acid) in the N6 medium showed improved induction of type-I calli over 1.0, 2.0, and 3.0 mg L^−1^ 2,4-D (Figure 2E). With this 0.50 mg L^−1^ 2,4-D concentration, the total callus induction rate reached as high as 75% on average (Figure 2F). After initial induction of the calli, they were sub-cultured on the media containing either 7 g agar/L or 3.0 g phytagel/L. The subculture medium with agar was more conducive to keep the growth of type-I calli with a higher induction rate (62.49%), as compared to that containing phytagel, although that use of agar and phytagel respectively in the initial callus induction did not significantly impact induction rates (74.44% and 71.67%) for the type-I calli (Figure 2G). Furthermore, an air drying (for 3.0 h) of the medium surface before the subculture could promote the conversion of the type-II calli into type-I calli (93.89%) (Figure 2H), which were suitable for the generation of embryogenic calli.

### 2.3. Effects of Regulators on Agrobacterium-Mediated Infection and Regeneration of Goosegrass Calli

To evaluate the infection efficiency of goosegrass calli, the empty plasmid with the *GUS* gene was transformed into the *Agrobacterium tumefaciens* strain (EHA105). First, the effect of *Agrobacterium*-mediated infection was determined by the *A. tumefaciens* cell culture with OD_600_ values from 0.40 to 0.80. The results showed that the optimal cell density for infection was 0.4–0.5 (OD_600_), which obtained the maximum transformation efficiency (39.17%) of the *GUS* gene on the goosegrass calli (Figure 3A). Secondly, we investigated the infection time of the goosegrass calli dipped with the *A. tumefaciens* culture for various time intervals from 10 to 40 min. The infection duration of 30 min was found to be optimal with 37.50% of the calli showing the *GUS* expression (Figure 3B). Thirdly, goosegrass calli were co-cultivated with *A. tumefaciens* cells for different periods from 0 to 4 days. The transformation efficiency of *GUS* was significantly improved with 2–3 days of co-cultivation, reaching as high as 36% (Figure 3C). On the other hand, different concentrations of acetosyringone (AS) with 100–500 μmol/L in the co-cultivation were tested. The results showed that use of 300 μmol/L AS produced the highest efficiency (58.67%) of goosegrass transformation (Figure 3D). Next, the optimal concentration of hygromycin (0, 20, 25, 30, 40, 50, 100 mg L^−1^) was determined for screening goosegrass transformants with the selectable marker gene *HPT* (encoding hygromycin phosphotransferase). We observed that the use of 25 mg L^−1^ of hygromycin dramatically decreased the fresh weight of the goosegrass calli (Figure 3E). Thus, the concentration of 20 mg L^−1^ of hygromycin was used to select the transformant cells during the callus culture. In addition, the use of a suitable antibiotic, such as timentin, for *A. tumefaciens* counter selection is crucial during the callus culture, which can avoid contamination of the *A. tumefaciens* cells over the surface of the calli. A test showed that the use of 100 mg L^−1^ timentin could drastically decrease the contamination rate of *A. tumefaciens* on the surface of the goosegrass calli (Figure 3F). Thus, this concentration was used for later goosegrass transformation.

For callus regeneration in plant transformation, MS or 1/2 MS medium is often used. Thus, the selected hygromycin-resistant goosegrass calli were transferred to MS medium supplemented with different combinations of 6-BA (0.50–2.00 mg L^−1^), 2,4-D (0.50 mg L^−1^), and kinetin KT (0.50–2.00 mg L^−1^) for regeneration (Table 1). The results showed the highest regeneration rate (71.05%) on MS medium supplemented with 0.50 mg L−1 6-BA and 0.50 mg L−1 KT.

### 2.4. Goosegrass Transformed with the Paraquat-Resistant Gene EiKCS

After establishing the the *Agrobacterium*-mediated genetic transformation system in goosegrass, we applied this system to study the gene function of *EiKCS* that confers paraquat-resistance in goosegrass [29]. Type-I calli were induced from seeds of the paraquat-susceptible goosegrass biotype (MZ04) on N6 medium with 0.50 mg L^−1^ 2,4-D (Figure 4A). After callus subculture (Figure 4B), the calli were infected by *A. tumefaciens* cells containing the binary vector pCUbi1309::*EiKCS* for overexpressing *EiKCS* driven by the promoter of the maize *Ubiquitin* gene (Figure 4C). Then, the calli were successively transferred to co-cultivation media with 0.50 mg L^−1^ AS and screening media with 20 mg L^−1^ hygromycin (Figure 4D). Next, the hygromycin-resistant calli were differentiated on MS with 0.50 mg L^−1^ 6-BA, 0.50 mg L^−1^ KT, 0.05 mg L^−1^ NAA, 0.05 mg L^−1^ IAA, and 100 mg L^−1^ timentin, until buds appeared and then they were transferred to light until green leaves grew to 1–2 cm in length (Figure 4E). The transformants were separately transferred to root-inducing media until green leaves grew to 4–6 cm in length (Figure 4F). Finally, acclimatization and soil transfer of the transgenic plants were carried out (Figure 4G). Each independent transformant was propagated into more than 14 plants in pots (Figure 4H).

### 2.5. Characterization of Paraquat-Resistant Transgenic Goosegrass Plants

To determine the paraquat-resistance and genetic characters of the transgenic goosegrass plants, they (T_2_ to T_5_ generations) were treated with the recommended paraquat solution (3750 mg L^−1^). After spraying with a full dose of the paraquat solution, all leaves of the transgenic (T_2_, T_3_) and wild-type plants withered within two days. However, the transgenic plants survived after 30 days and almost recovered after 90 days, whereas the wild-type plants completely died (Figure 5A). Moreover, the abiotic stress of paraquat stimulated the growth of transgenic goosegrass. Although the leaves of the transgenic goosegrass (T_4_) were visibly damaged after two days of full-dose or half-dose paraquat treatments, and their vegetative growths were still flourishing to the reproductive stage after 30 days of the treatments (Figure 5B). In addition, the paraquat resistance of the transgenic goosegrasses (T_5_) was equivalent to that of naturally selected resistant-biotype goosegrass QY05. However, the susceptible-biotype goosegrass (JM) as a control died under a low concentration of 270 mg L^−1^ paraquat treatment (Figure 5C). Hence, these results determined that a stable genetic resistance to paraquat could be successfully obtained by using this *Agrobacterium*-mediated genetic transformation system in goosegrass.

To confirm the positive transgenic goosegrass, the T_2_ to T_6_ generations of transgenic goosegrass were detected by molecular screening. PCR certified positive transgenic lines in T_2_ to T_6_ generation goosegrasses with specific primers. The amplifying fragments contained the *EiKCS* gene with the Ubiquitin-promoter sequences in the over-expression vector. The presence of the marker *hygromycin* (*HPT*) gene and Ubi were the positive controls. And the original paraquat-susceptible goosegrass biotype (MZ04) was used as the negative control (Figure 6A). The result showed the successful identification of positive transgenic goosegrass lines. Meanwhile, the effect of the *EiKCS* function was evaluated in the paraquat-resistant response of the transgenic goosegrass. The qRT-PCR showed different expression levels of *EiKCS* in leaves of the wild-type (MZ04) and three OE-*EiKCS* lines (T_2_) under treatments of spraying H_2_O, 270 mg L^−1^ paraquat, and 270 mg L^−1^ paraquat plus 1.5 mmol/L spermidine (Figure 6B). The results of qRT-PCR confirmed that the relative expressions of the *EiKCS* gene in transgenic goosegrass were all higher than those in wild-type goosegrass. Moreover, paraquat stress promoted significant overexpressing of *EiKCS* in these transgenic lines with an average of 7.38-fold compared to the wild-type goosegrass. However, the increased expression of *EiKCS* by paraquat in the transgenic goosegrass was down-regulated by 4.25-fold when applying exogenous spermidine (Para + spd). Moreover, the gene-targeted protein EiKCS showed higher expression levels in the transgenic goosegrass than in all wild-type goosegrass (MZ04) by PRM under the same treatments (Figure 6C). The results of PRM confirmed the relative abundance of protein EiKCS in these transgenic lines with a 1.16-fold (H_2_O), 1.10-fold (Para), and 1.32-fold (Para + spd) increase compared to the wild-type goosegrass on treatments. The quantification methodology showed that the total expressed mRNA and protein of the *EiKCS* in transgenic goosegrass were higher than the endogenous expression of those in wild-type goosegrass. Therefore, this transgenic goosegrass was confirmed as an engineering plant for overexpressing *EiKCS*.

With these treatments, the resistance phenotypes of the transgenic goosegrass were consistent with the relative expression levels of *EiKCS* (Figure 6D). The paraquat-susceptible goosegrasses (MZ04) thoroughly died, while transgenic goosegrass survived with green leaves under the same paraquat stress. Moreover, the pre-treatment with exogenous spermidine significantly alleviated the toxicity of paraquat, making the stems and leaves of the plant green again. The fresh weights of goosegrass were statistically analyzed at 48 h after these treatments (Figure 6E). The results showed that paraquat-susceptible goosegrasses were more significantly inhibited by paraquat, while the transgenic grass improved its tolerance to paraquat. The high expressions of the targeted *EiKCS* and its translated proteins positively affect the paraquat resistance of goosegrass. Meanwhile, the exogenous polyamine inhibited the level of the *EiKCS* expression, indicating that the EiKCS might involve in polyamine metabolism. The results are generally consistent with those of rice, but whether the same biosynthesis pathway of polyamines between goosegrass and rice still requires further study on this transgenic material. In summary, this effective *Agrobacterium*-mediated genetic transformation system in goosegrass is a beneficial biotechnological approach for exploring the gene functions of engineering weed species in response to abiotic stresses.

## 3. Discussion

Given the successful *Agrobacterium*-mediated genetic transformation in important crops, such as rice [30,31] and maize [32], this approach has been widely applied in many plant species. The breeding of transgenic herbicide-resistant crops has been drastically increased to address the limitations posed by weed damage and improve crop productivity [33,34,35]. However, further research is needed to identify genes with multiple resistances for creating herbicide-resistant crops and to explore the molecular mechanism of non-target genes in herbicide detoxification [36,37]. In our previous studies, the mechanism of paraquat resistance was considered to involve non-target genes [27]. The *EiKCS* was found in goosegrass with not mutated in the resistant biotype (R-NX) compared to the susceptible biotype (S-HN). Further, the *EiKCS* has been shown as an ideal candidate gene of the paraquat-resistant genes for transgenic rice by promoting the polyamine synthesis involved with the arginine decarboxylase (ADC) pathway [29]. In this study, we transformed goosegrass for overexpression of *EiKCS*. We found that the transgenic goosegrass showed a more distinct paraquat-resistance based on visual injury assessment, as compared to the transgenic rice. However, the new molecular mechanism responses (ADC pathway or not) of the polyamine pathway regulated by the *EiKCS* gene to paraquat resistance could be further elaborated in further metabolic research of this transgenic goosegrass. Thus, analyses of paraquat-resistance phenotype, stable inheritance, and expressions of the transgenes showed that this *Agrobacterium*-mediated transformation method should be more useful in studying gene function and the related metabolism pathways in this weed species.

In plant *Agrobacterium*-mediated transformation processes, the problems such as bacterial contamination of seeds for callus induction, hydration and browning of calli are not uncommon. In this study, we established the first efficient *Agrobacterium*-mediated transformation system in goosegrass by use of a combination of optimal factors including explants and regulators on the callus induction and culture, *Agrobacterium*-mediated infection, and callus regeneration. It is worth mentioning that the contamination of QY04 and PY07 calli came from their seeds at rates of 11.83% and 21.89%, compared to almost no contamination in MZ04 under the same treatment of seed-sterilizing. This difference in the contamination rate of seeds may be related to their originally collected geographical locations. Second, primary calli were easy to hydrate in the process of the subculture. We found that in the subculture medium, the use of 7 g agar per L instead of 3 g phyto gel per L could effectively generate type-I calli to promote the growth of embryogenic callus, but did not affect the formation of type-I calli in the callus induction medium, possibly due to that use of 7 g agar in the subculture decreased the hydration degree of calli. Next, the browning of the calli also was caused by the growth of *Agrobacterium* cells on the surfaces of the calli after the infection. It is best to adjust the cell OD_600_ to 0.5 during the infection, but it also is recommended to add an antibiotic (such as 100 mg L^−1^ timentin as used in this study) when the surfaces of the calli are contaminated severely by *Agrobacterium* cells. Thus, it is clear that we performed a thorough assessment of the methods and attempted to explain the reasons for testing certain components, how they were evaluated, and what the best recommendation should be.

In conclusion, this is the first establishment of an *Agrobacterium*-mediated genetic transformation in goosegrass, which is novel and has value in future research using similar methods for herbicide resistance research.

## 4. Materials and Methods

### 4.1. Plant Materials

The experiments were carried out at Weed Research Laboratory, South China Agricultural University, Guangzhou City, Guangdong Province, China. Natural seeds of four paraquat-susceptible biotypes (MZ04, PY07, JM and QY04) and one paraquat-resistant biotype (QY05) were selected from 16 biotypes of mature goosegrass in Guangdong province, which were MZ04, PY07, JM, QY04, and QY05 with their paraquat GR_50_ values of 24.29, 54.15, 66.11, 67.29 and 314.43 g a.i.ha^−1^ respectively [38]. Among 16 biotypes of goosegrass, the MZ04 biotype was the most susceptible to paraquat (GR_50_ 24.29 g a.i.ha^−1^), while the QY05 biotype (GR_50_ 314.43 g a.i.ha^−1^) was the most resistant. Mature seeds of the MZ04 biotype goosegrass were used for the *Agrobacterium*-mediated transformation with *EiKCS* in this study. Paraquat-susceptible seeds of goosegrass (S-HN) and paraquat-resistant seeds of goosegrass (R-NX) in Guangdong Province of China were also used in determining developmental regulators in the genetic transformation system. The level of paraquat-resistance in the R-NX goosegrass was 59.48-fold higher than that in the S-HN goosegrass as previously described [27].

### 4.2. General Improvements of the Transformation System in Goosegrass

The genetic transformation system of goosegrass was improved by determining the developmental regulators. A semi-manufactured product of N6, MS, and 1/2 MS medium (Qingdao Hope Bio-Technology Co., Ltd., Qingdao, China) respectively used as the basal N6 as a calli-induction, differentiation, and root-induction medium for calli induction of MZ04 biotype goosegrass. The product of hygromycin and timentin (RealTimes, Bio-Tech Co., Ltd., Beijing, China) solved a contamination problem by inhibiting *Agrobacterium* overgrowth on the surface of calli. The X-Gluc kit (Coolaber, Beijing, China) was used to detect the transient transformation of *gus* in goosegrass calli by monitoring the presence of blue color. Other reagents in the study were purchased from Sigma-Aldrich^®^ Brand (Merck-Sigma, Darmstadt, Germany and/or its affiliates). The explant, variety, and developmental regulators of goosegrass were analyzed using more than 50 biological replicates. Different tissues of goosegrass (R-NX) such as seeds, stems, leaves, and roots were used as explants for callus induction. The effects of the developmental regulators were evaluated on the induction of calli from goosegrass seeds. To break the seed dormancy, the goosegrass seeds were respectively treated by ddH_2_O (S-HN), 30% NaOH (S-HN and R-NX), −20 °C (S-HN), 4 °C (S-HN) and 75% ethanol (S-HN), according to the previous reports [39,40]. Meanwhile, the factors of agar, phytagel, HgCl_2,_ 2,4-D, 6-BA, KT, IAA, NAA, OD_600_ of *Agrobacterium*, AS, hygromycin, timentin, and the drying all were tested in the calli of goosegrass (S-HN) in different stages of inducing, transforming and regenerating. And more than 20–30 seeds or 10–15 calli of goosegrass were placed on each dish (120 × 20 mm) of the different media with a minimum of 20 dishes per batch. The rate of induction or germination (%) = the number of calli or germination/the number of seeds × 100%. The induction rate of type I or calli contamination rate (%) = the number of target calli/the number of calli × 100%. The transformation efficiency of *gus* = the number of blue calli/the number of stained calli.

### 4.3. Vector Construction for Overexpressing EiKCS

Vector construction with the genetic map for overexpressing *EiKCS* was already performed in transgenic rice according to our previous studies, including the sequence of its translated protein EiKCS [29]. In brief, a paraquat-resistant gene (*EiKCS*/*PqE*) was cloned by PCR from R-NX and subsequently was constructed into a pCUbi1390 vector with the strong promoter of the maize *Ubiquitin* gene. The pCUbi1309::*EiKCS* construct was transferred into *Escherichia coli* DH5α and then introduced into *Agrobacterium* strain EHA105 by electroporation (1800 V/2 mm). Positive *Agrobacterium* colonies of pCUbi1309::*EiKCS* were screened using yeast extract broth (YEP) media with 50 mg mL^−1^ hygromycin and 20 mg mL^−1^ rifampicin [29]. Figure 7 shows a graphical overview of vector construction and transformation system for goosegrass.

### 4.4. Transformation System for Goosegrass

The EHA105 cells containing pCUbi1309*::EiKCS* were used to infect goosegrass calli (Type I) of the paraquat-susceptible biotype (MZ04). The *Agrobacterium*-mediated transformation process for goosegrass (MZ04) was divided into six steps. (1) Breaking seed dormancy. Mature goosegrass seeds of MZ04 were refrigerated at −20 °C for more than 30 days. (2) Seed sterilization. Seeds (MZ04) were soaked in 75% ethanol for 1 min, followed by surface sterilization using 0.05% HgCl_2_ solution for 10 min with stirring per 3 min. (3) Induction of calli. Each liter of callus induction medium was composed of 24.1 g basal N6, 0.5 g proline, 0.6 g hydrolytic casein, 10 g sucrose, 3 g phytagel, 1 mL 2,4-D (0.5 mg L^−1^), and with pH 5.8. Seeds (MZ04) were placed on the callus induction medium at 24 °C/24 h in darkness for 30 days. (4) Callus subcultures. Each liter subculture medium was composed of 24.1 g basal N6, 0.5 g proline, 0.6 g hydrolytic casein, 10 g sucrose, 7 g agar, 0.5 mL 2,4-D (0.5 mg L^−1^), and pH 5.8. The sponge tissue, buds, and seed coats attached to the calli needed to be removed. The newly formed pale yellow, compact, and nuclear embryogenic calli were transferred to the subculture dish containing the subculture medium at 24 °C/24 h in darkness for 60 days. After being transferred to new subculture dishes per 30 days, these calli grew to a diameter of about 1 cm. (5) Preparation of *A. tumefaciens* competent cells. The strain EHA105 containing the overexpression vector (pCUbi1309::*EiKCS*) was cultured in the 100 mL YEP media by 250 mL conical flask at 28 °C for 12 h at 230 rpm on an incubator shaker to obtain the OD_600_ at 0.5000–0.5999. (6) Callus infection, co-cultivation and screening. ① The dry granular calli with pale yellow (Type I callus) were pretreated at 4 °C for 1–2 days before in the 100 mL *Agrobacterium* solution completely dipped for 30 min. ② The calli spread over filter paper and covered with filter paper for 3 h air-dry, and then papers were replaced 4–5 times to absorb the excess bacterial solution. ③ The dry calli were transferred to a co-culture medium in 24 °C/24 h darkness for three days. The co-culture medium was the induction medium supplemented with 300 μmol/L acetosyringone (AS) and pH 5.2. ④ Then, the calli were vigorously shaken for 30 s followed by standing for 5 min each time until the ddH_2_O became clear. This rinse was needed to change the ddH_2_O around 4–5 times. The calli were dipped for 30 min in a ddH_2_O supplemented timentin (100 mg L^−1^), and they were repeatedly filtered as in ② after discarding this ddH_2_O. ⑤ The clean and dry calli were transferred to an A-screening medium for 25 days and then to a B-screening medium for 20 days. The A-screening medium was the induced medium supplemented with hygromycin (15 mg L^−1^), carbenicillin (200 mg L^−1^) and timentin (100 mg L^−1^). The B-screening medium was the induced medium supplemented with hygromycin (20 mg L^−1^), carbenicillin (300 mg L^−1^) and timentin (100 mg L^−1^). All screening media altered 3 g phytagel to 7 g agar and pH to 6.0. During the screening period, the calli needed to be rinsed as in ④, dried as in ②, and transferred to a corresponding screening medium when *A. tumefaciens* appeared on the surface of the calli.

### 4.5. Regeneration System of Goosegrass Calli

The regeneration system of the goosegrass (MZ04) transformant calli after hygromycin screening was divided into four steps. (1) Differentiation of transformant calli. The differentiation medium was composed of 41.74 g basal MS with 6-BA (0.5 mg L^−1^), KT (0.5 mg L^−1^), NAA (0.05 mg L^−1^), IAA (0.05 mg L^−1^), timentin (100 mg L^−1^) on a per L basis, pH 5.7 ± 0.1. Transformants of goosegrass were placed in 24 °C/24 h darkness for more than 15 days until leaves appeared. Then, transformants with leaves were transferred to 24 °C/14 h light (80 μmol/m^2^·s) and 10 h darkness per day for more than 40 days until green leaves grew to 1–2 cm in length. (2) Root induction of transformants. The root induction medium was composed of 39.45 g basal 1/2 MS with hygromycin (20 mg L^−1^) and timentin (100 mg L^−1^). The transformant calli were placed in root induction conical flasks with 24 °C/14 h light (80 μmol/m^2^·s) and 10 h darkness until green leaves grew to 4–6 cm in length. (3) Transgenic plants were acclimated for the field. Transgenic plants were transferred into sterilized nutrient soil with sterile water mixed at the ratio of 1:1 with the surface compacted and moist. The transgenic goosegrass plants were grown in a greenhouse at 34 °C/28 °C (day/night) for 12 h in each day. (4) Single-plant propagation of transgenic goosegrass in the greenhouse. Transgenic goosegrass (T_0_ plants) was self-pollinated to produce the T_1_ generation, and T_2_ generation was molecularly screened from the T_1_ generation segregation under paraquat stress, and so on.

### 4.6. Molecular Screening of Transgenic EiKCS Goosegrass

Genomic DNA was extracted from the leaves of transgenic goosegrass (OE-*EiKCS*) by TIANcombi DNA Lyse&Det PCR Kit. The presence of the *EiKCS* gene in transgenic goosegrass (T_0_ and T_1_ generations) was detected by primer Ubi-Tnos-F/R (or pCUbi1390-F/R) 5’-TTTAGCTCATACG-3’/5’-TTGCGGGACTATCATAA-3’, which amplified the sequence (1898 bp) from the *Ubiquitin*-promoter to Tnos in the vector consisting of the *EiKCS* sequence. The presence of *EiKCS* gene in transgenic goosegrass (T_2_, T_5_ and T_6_ generations) was detected by Ubi-EiKCS-F/R (or KCS_1_-F/R) primer 5′-CCTGCCTTCATACGCTATTT-3′/5′-ATCTTGCGCTGAAAGTCC-3′, which amplified the sequence (450 bp) from the vector to a section of *EiKCS* gene sequence including *Bam*HI digestion site. Moreover, the full-length of the *EiKCS* (1551 bp) and the *EiKCS* with part sequence of its original promoter (1067 bp) were separately detected by the primer *Full-EiKCS-F/R* 5′-ATGGACAACCCCGCGGCGCCGAGCAAT-3′/TCATTCGCTG GAAAGCTTGGAAACCT and the Pro-*EiKCS*-F/R 5′-GCTAAGTAGGAGGAGGCGGTGT TAT-3′/5′-GGATGCCGATGTCCTTGGGCTTCA-3′. The presence of the marker *hygromycin* (*HPT*) gene and *Ubiquitin*-promoter were positive controls. *HPT* was detected by the primer Hyg-F/R 5′-ACGGTGTCGTCCATCACAGTTTGCC-3′/5′-TTCCGGAAGTGCTTG ACATTGGGGA-3′, which amplified sequence (509 bp) consisted of the vector sequence and the *hygromycin* (289 bp), and by the primer HPT2-F/R 5′-GTGCTTGACATTGGGGA GTT-3′/5′-ATTTGTGTACGCCCGACAGT-3′ was used for amplification (698 bp). The primer Ubi-F/R 5′-CTACCTTCTCTAGATCGGCGTT-3′/5′-CGTATGAAGGCAGGGCTAA A-3′ amplified the *Ubiquitin*-promoter. The original MZ04 goosegrass biotype was used as the negative control.

### 4.7. Real-Time Quantitative RT-PCR

Three lines of transgenic goosegrass (T_2_ generation) and paraquat-susceptible goosegrasses (MZ04) were planted in pots with each pot containing five seedlings. They were sprayed by a 3WP-2000 spray tower (Nanjing Research Institute for Agricultural Mechanization, Ministry of Agriculture, Nanjing, China) with 270 mg L^−1^ paraquat (Syngenta Corporation, Shanghai, China), sprayed with 270 mg L^−1^ paraquat 12 h after spraying with 1.5 mmol/L spermidine (Merck-Sigma, Darmstadt, Germany and/or its affiliates), or sprayed with ddH_2_O. After 48 h, the collected transgenic goosegrass lines of OE_1_, OE_2_, and OE_3_ were named OE-P-1, OE-P-2, OE-P-3, OE-P-S-1, OE-P-S-2, OE-P-S-3, OE-H_2_O-1, OE-H_2_O-2, and OE-H_2_O-3. Meanwhile, W_1_, W_2_, and W_3_ of MZ04 goosegrasses indicated wild type as the negative control. Then, leaves of these samples were frozen in liquid nitrogen and stored at −80 °C.

For real-time quantitative RT-PCR (qRT-PCR), cDNA sequences of the *EiKCS* gene (144 bp) and the internal marker *Actin* gene (150 bp) from the leaves of transgenic goosegrass and MZ04 goosegrass biotype were amplified by the primers KCSP-F/R 5’-CAAGGTGCTCAAGCGGAAA-3’/5’-GGCTCCATGTGCCAGTCC-3’ and the primers Actin-F/R 5’-GACGAGTCTGACCCATCCATT-3’/5’-GTTGAAAACTTTGTCCACGCTA- 3’ to calculate the relative expression of the *EiKCS* transcripts. Total RNAs were extracted from leaves of both transgenic *EiKCS* goosegrass and MZ04 goosegerass by Unlq-10 column Trizol total RNA Extraction Kit (Sangon Biotech, Shanghai, China). Then the cDNAs were synthesized by Maxima Reverse Transcriptase (Thermo Scientific, Waltham, MA, USA). A 2 μL aliquot of cDNA template was used with SybrGreen qPCR Master Mix (Sangon Biotech, Shanghai, China). The qRT-PCR was carried out using a StepOne Plus instrument of (ABI, Foster, CA, USA). The reactions were performed under the conditions of 95 °C 3 min, 45 cycles for 95 °C 5 s, 60 °C 30 s, followed by 95 °C 15 s, 60 °C 60 s and 95 °C 15 s for dissociation curve analysis.

### 4.8. Parallel Reaction Monitoring

The PRM (Parallel Reaction Monitoring) was based on targeted quantitative proteome technology that used mass spectrometry (MS) to identify the target protein. The protein EiKCS was detected by PRM using its unique peptide segments (SGLGEETYLPAAVLR and CFGCVTQEEDGEGR of EiKCS) in the same samples of RT-PCR, which were three biological replicates. The method of PRM detection was previously reported [41].

### 4.9. Paraquat-Resistance Analysis of Transgenic EiKCS Goosegrass

The different lines of transgenic goosegrass (T_2_ to T_6_ generation) were sprayed at the vegetative and reproductive stages with 3750 mg L^−1^ paraquat (Gramoxone Max 2.5 pts/a, Syngenta, Basel, Switzerland) as the recommended dose in the field. The MZ04 biotype goosegrass was the control. The nine T_2_ transgenic goosegrasses were sprayed with full-dose paraquat. The 15 T_3_ transgenic goosegrasses from seeds of the same T_2_ line (12-1-1) were sprayed with full-dose paraquat. Moreover, the plants of T_4_ trangenic goosegrass in 28 pots from seeds of the same T_3_ line (12-1-1-2) were sprayed with full-dose and half-dose paraquat. The 9 pots of them were sprayed with full-dose paraquat, and 3 pots were without spraying. And the 12 pots of them were sprayed with half-dose paraquat, and 4 pots were without spraying. Furthermore, the 24 seedings at the 4–6 leaves stage of T_5_ transgenic goosegrass from the seeds of T_4_ lines (12-1-1-2-8) were sprayed with full-dose paraquat. T_5_ transgenic goosegrass planted in 4 pots was selected from the seeds of two T_4_ lines (12-1-1-2-8 and 12-1-1-2-28) with each pot of four plants. The 3 pots of them were sprayed with 270 mg L^−1^ paraquat (Syngenta, Shanghai, China) and 1 pot without spraying, keeping the same treatment as the transgenic rice [20]. The paraquat-susceptible goosegrass (JM) and paraquat-resistance goosegrass (QY05) were treated similarly for comparison. In addition, the T_6_ transgenic goosegrass lines from the seeds of T_5_ lines (12-1-1-2-28-3) were sprayed with 270 mg L^−1^ paraquat (Syngenta, Shanghai, China) or sprayed with 270 mg L^−1^ paraquat 12 h after spraying with 1.5 mM spermidine, keeping the same treatments as the transgenic rice [20]. The paraquat-susceptible goosegrass (MZ04) and paraquat-resistance goosegrass (QY05) were treated similarly for comparison. Photographs were taken before and after paraquat treatments to record the resistant phenotypes of complete death or survival with green leaves in goosegrass. The seeds of T_6_ transgenic goosegrass, MZ04, and QY05 were simultaneously sown and transplanted. Although MZ04 and QY05 grew better than T_7_ transgenic goosegrass, their fresh weights were calculated 48 h after the same treatments.

### 4.10. Data Analysis

Significant differences were analyzed by analysis of variance (ANOVA) and Duncan’s test. Figures were created using DPS 7.05 (Hangzhou, China), OriginPro 8.5.0 (OriginLab Corporation, Northampton, MA, USA), GraphPad Prism 8.3.0 (GraphPad Software, LLC, San Diego, CA, USA), and Adobe Illustrator CS4 (Adobe Systems Incorporated, San Jose, CA, USA).

## Figures and Tables

**Figure 1 ijms-24-06629-f001:**
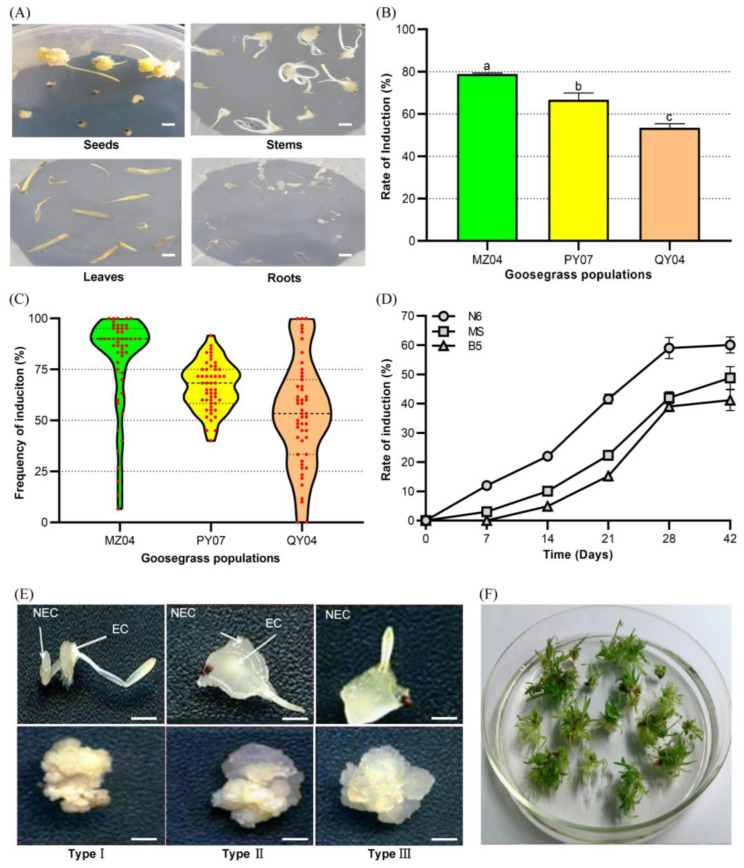
The callus induction for goosegrass. (**A**) Effects of the seeds, stems, leaves, and roots of goosegrass as explants for callus induction. Scale bars, 0.5 cm. (**B**) The average efficiency of callus induction of seeds from three paraquat-sensitive goosegrass biotypes. Different letters indicate significant differences at *p* < 0.05. (**C**) The distribution of frequency of callus induction from seeds of the three biotypes. (**D**) The efficiency rates of callus induction from MZ04 seeds in N6, MS, and B5 media. (**E**) The morphology of the primary calli of the types I to III recognized by the texture. EC means embryogenic calli and NEC non-embryogenic calli. Scale bars, 0.5 cm. (**F**) The high-efficiency differentiation was from the types-I calli of MZ04.

**Figure 2 ijms-24-06629-f002:**
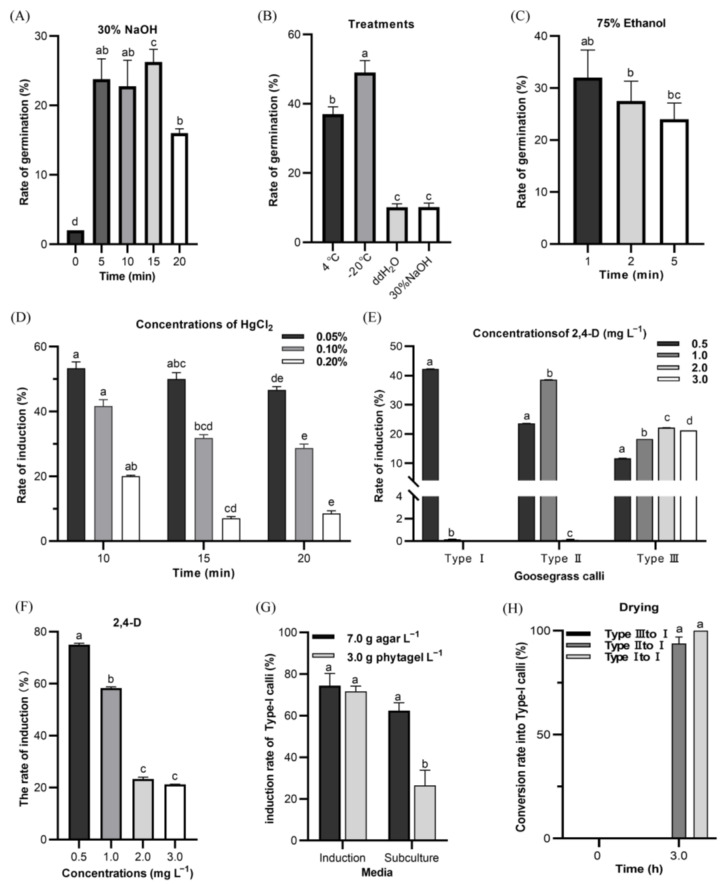
The optimal conditions for seed gemination and callus induction of goosegrass. (**A**) The germinating rates of seeds in different treatment time of 30% NaOH solutions. Non-treatment at 0 min as the control. (**B**) The germinating rates of seeds in various treatments of 30% NaOH for 15 min, ddH_2_O for 24 h, 4 °C or −20 °C for about 30 days. (**C**) The germinating rates of seeds in different treatment time of 75% ethanol (and then with 0.10% HgCl_2_ for 15 min). (**D**) The rates of seed inducing after the treatments of different concentrations of HgCl_2_ in 75% ethanol solution. (**E**) The induction rates of the types-I to -III calli in induction media with different concentrations of 2,4-D solution. (**F**) The average of callus induction rates in induction media with different concentrations of 2,4-D solution. (**G**) The induction rates of the type-I calli in induction and subculture media with 7.0 g agar/L and 3.0 g phytagel/L, respectively. (**H**) The conversion rates of the type-II and type-III calli into type-I calli in subculture with an air drying of the medium surface for 0 h and 3.0 h, respectively, before the subculture. Different letters indicate significant differences at *p* < 0.05.

**Figure 3 ijms-24-06629-f003:**
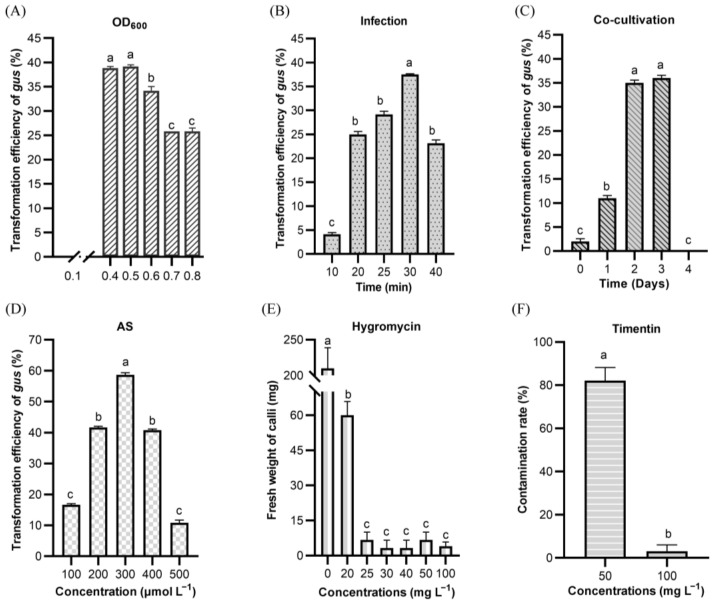
The determination of optimal regulators by *Agrobacterium*-mediated infection of goosegrass calli. (**A**) The effects of OD_600_ of *A. tumefaciens* on the goosegrass calli showing *gus* expression. (**B**) The impact of infection time on the goosegrass calli showing *gus* expression. (**C**) The effects of co-cultivation time on the goosegrass calli showing *gus* expression. (**D**) The effect of different acetosyringone (AS) concentrations in the co-cultivation on the transformation efficiency of the goosegrass calli. (**E**) Determination of hygromycin concentration for selecting *HPT*-positive goosegrass calli. The treatment without hygromycin (0) served as a negative control. (**F**) The effect of timentin as a suitable antibiotic for *A. tumefaciens* counter selection on goosegrass calli. Different letters indicate significant differences at *p* < 0.05.

**Figure 4 ijms-24-06629-f004:**
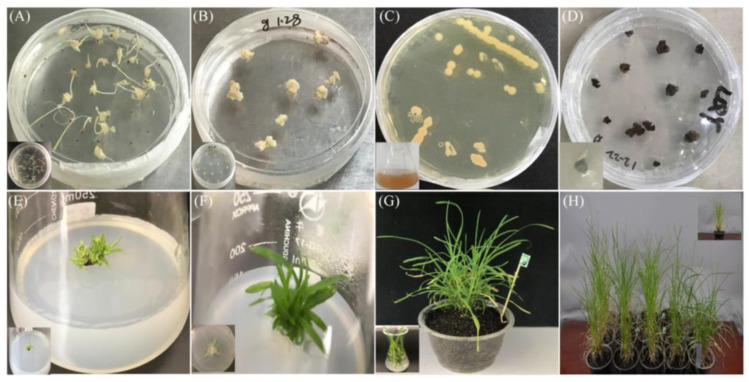
*Agrobacterium*-mediated genetic transformation of the *EiKCS* gene in goosegrass. (**A**) Callus induction with MZ04 seeds. (**B**) Callus subcultures. (**C**) Preparation and preservation of *A. tumefaciens* cells with the pCUbi1309::*EiKCS* vector. (**D**) Infection, co-cultivation and screening of calli. (**E**) Differentiation of transformants. (**F**) Root induction of transformants. (**G**) Acclimatization and field transfer of transgenic plants. (**H**) Propagation of transgenic goosegrass plants.

**Figure 5 ijms-24-06629-f005:**
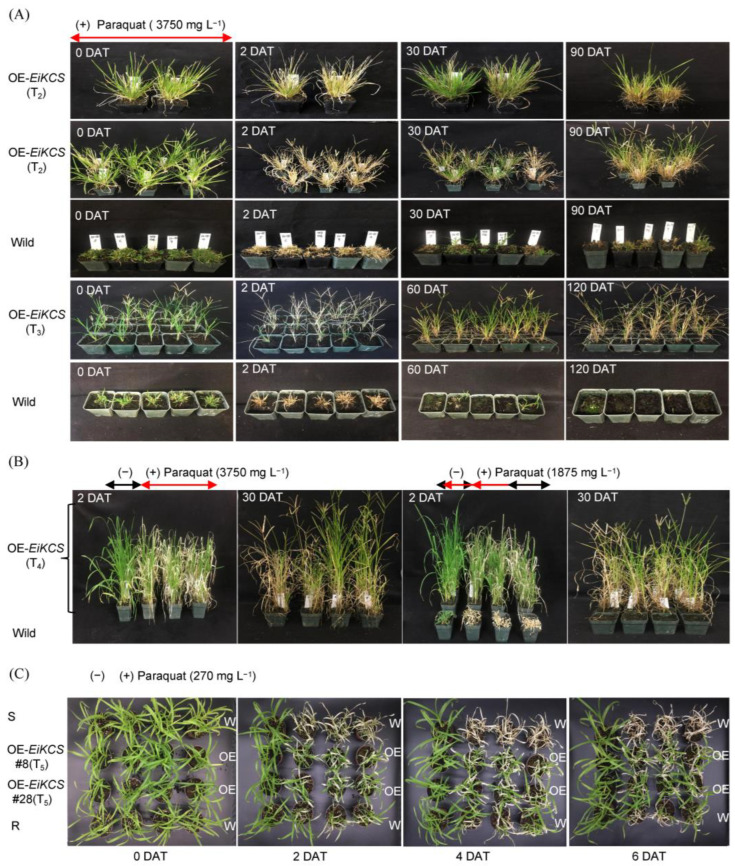
Paraquat resistance of the transgenic goosegrass plants. (**A**) The resistant phenotypes of transgenic goosegrass (T_2_ and T_3_ generation) at the vegetative or reproductive stage in 120 days after spraying 3750 mg L^−1^ paraquat. Wild type means the susceptible goosegrasses (MZ04) used as the control. (**B**) The resistant phenotypes of transgenic goosegrass (T_4_ generation) at reproductive stage in 30 days after spraying 3750 mg L^−1^ and 1875 mg L^−1^ paraquat. Wild type means the susceptible goosegrasses (MZ04) used as the control. (**C**) The resistant phenotypes of the transgenic lines line-8 and line-28 (T_5_ generation) after treatment with 270 mg L^−1^ paraquat. R means a naturally selected resistant-biotype goosegrass (QY05), and S means the susceptible goosegrasses (JM) used as the control. W means wild-type goosegrass of natural evolution, and OE means the overexpressing *EiKCS* of transgenic weeds. DAT means days of treatment.

**Figure 6 ijms-24-06629-f006:**
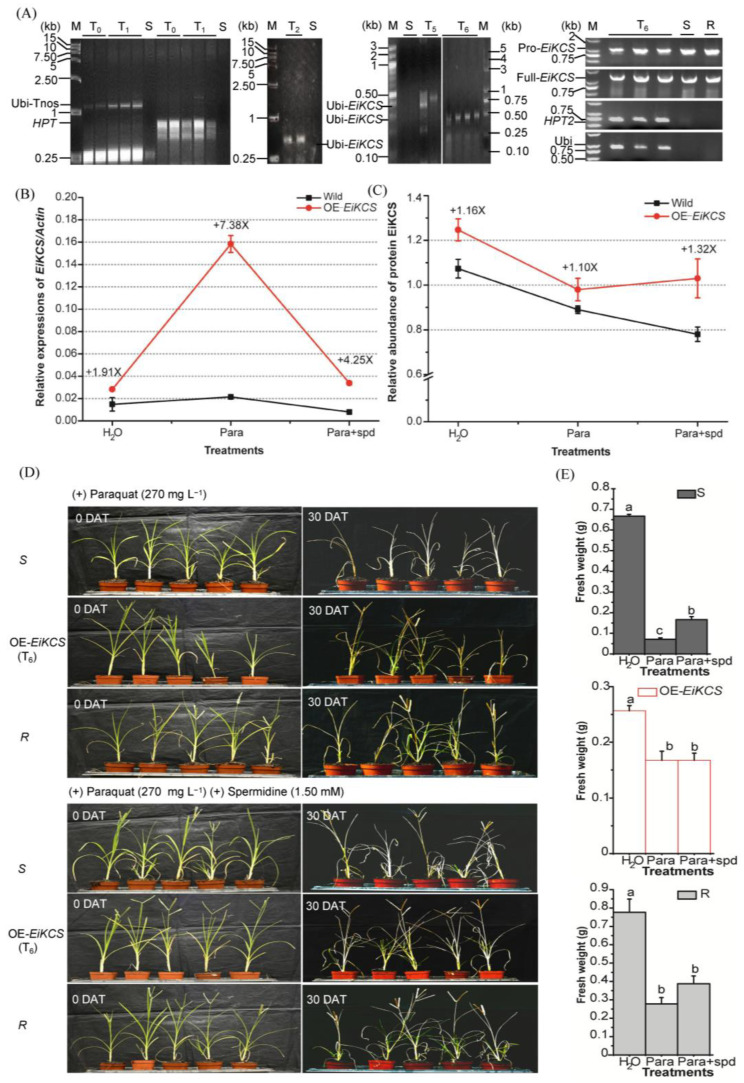
Molecular characterization of *EiKCS* on the paraquat-resistant response of the transgenic goosegrass. (**A**) Stable inheritance of the transgenes in different generations (T_0_, T_1_, T_2_, T_5_, and T_6_) assayed by PCR. S and R indicate naturally selected the paraquat-susceptible (MZ04) and paraquat-resistant (QY05) goosegrasses. Amplification fragments of the Ubi to Tnos (1.89 kb), the Ubi to a part of *EiKCS* (0.45 kb), the original-promoter to a part of *EiKCS* (1.07 kb), the full length of *EiKCS* (1.55 kb), The part of Ubi (0.91 kb). And *HPT* (0.53 kb) and *HPT2* (0.70 kb) used as control. (**B**,**C**) Relative expressions of the gene *EiKCS* in transgenic goosegrass by qRT-PCR under different treatments with *Actin* as an internal control (**B**). Relative abundance of the protein EiKCS in transgenic goosegrass by PRM under different treatments (**C**). Wild and OE-*EiKCS* means the susceptible goosegrass (MZ04) and the transgenic goosegrass plants (T_2_), respectively. The treatments included spraying H_2_O (H_2_O), 270 mg L^−1^ paraquat (Para), and 1.5 mmol/L spermidine (Spd). (**D**) The paraquat-resistant phenotypes of the transgenic goosegrass plants under different treatments of 270 mg L^−1^ paraquat, and 1.5 mM spermidine. R means resistant biotype (QY05), and S means susceptible biotype (MZ04). (**E**) Fresh weights of the transgenic goosegrass plants were assessed 48 h after the same treatment of H_2_O, paraquat, or spermidine. R means resistant biotype (QY05), and S means susceptible biotype (MZ04). DAT means days of treatment. Each had more than three replicates. Different letters indicate significant differences at *p* < 0.05.

**Figure 7 ijms-24-06629-f007:**
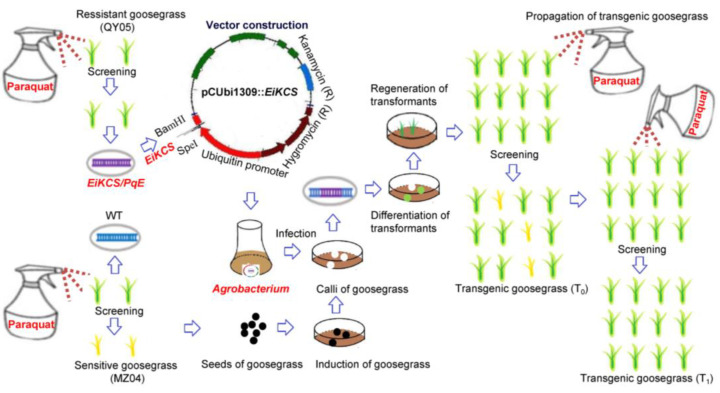
The graphical overview of the vector construction and transformation system for goosegrass.

**Table 1 ijms-24-06629-t001:** The optimal determination of regulators on regeneration of goosegrass calli.

Regulators	Concentration (mg L^−1^)	Regeneration Rate (%)
		Means ± SD
6-BA	0.50	0 ± 0 d
6-BA	1.00	0 ± 0 d
6-BA	2.00	0 ± 0 d
6-BA + 2,4-D	0.50 + 0.50	0 ± 0 d
6-BA + 2,4-D	1.00 + 0.50	0 ± 0 d
6-BA + 2,4-D	2.00 + 0.50	0 ± 0 d
6-BA + KT	0.50 + 0.50	71.05 ± 0.18 a
6-BA + KT	1.00 + 1.00	58.03 ± 0.08 b
6-BA + KT	2.00 + 2.00	50.02 ± 0.14 c

Note: Each treatment contained 30 calli of three replicates. Different letters indicate significant differences at *p* < 0.05. 2,4-D (2,4-Dichlorophenoxyacetic acid); 6-BA (6-Benzylaminopurine); KT (Kinetin).

## Data Availability

The data presented in this study are available on request from the corresponding author. The data are not publicly available due to protecting parts of the data by a Chinese Patent (ZL202110962351.0).

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
