# Peer review of "Establishment of an Efficient Agrobacterium-Mediated Genetic Transformation System to Enhance the Tolerance of the Paraquat Stress in Engineering Goosegrass (Eleusine Indica L.)"

_ijms, 2023, doi:10.3390/ijms24076629_

Round 1
Reviewer 1 Report
In the present manuscript, Luo and colleagues worked on an Agrobacterium-mediated genetic transformation system to enhance the tolerance of the paraquat stress in engineering goosegrass. The study was a straight-forward research. The research scientifically sound and the results are understandable.
Some comments are as below:
- It is recommended not to use the active form of the verb in scientific articles. Please correct this issue in the abstract and whole manuscript.
- Please add keywords such as Agrobacterium, goosegrass, genetic transformation, etc. to broaden the visually of the article for various readers.
- Abstract is focused on results and did not provide a good overview of the goal and final conclusion. It is highly recommended to revise this important part.
- In the title of the manuscript paraquat is given but in the introduction less details is provided. Please revise the introduction based on this issue.
- Introduction is too short and it is highly recommended to add more details and background related to the done research.
- Please provide a graphical overview of the vector construction and transformation system for goosegrass.
- Materials and methods section is too long and in some parts it looked like a result. It is recommended to revise this section and use appendix for extra information.
Author Response
RESPONSE TO REVIEWERS
of “ Establishment of an Efficient Agrobacterium-Mediated Genetic Transformation System to Enhance the Tolerance of the Paraquat Stress in Engineering Goosegrass (Eleusine indica L.)”
by Qiyu Luo, Shu Chen, Hai Nian, Qibing Ma, Yuyao Ding, Qinwen Hao, Jiping Wei, Jinesh D. Patel, Joseph Scott McElroy, Yaoguang Liu , and Yong Chen,
21th of March of 2023
We would like to thank you for your kind review of the manuscript. Your inputs are very helpful for improving the manuscript. We agree with almost all your comments and we have revised our manuscript accordingly.
We respond to each of the comments in details below . We are already crafting a revised version of the paper that it states the background of our work in the introduction more sufficient than before. Moreover, we are including all your suggestions and clarifying the text when needed. We hope that you will find our responses to your comments satisfactory.
Please, find below the your comments repeated in italics and our responses inserted after each comment. In the requirement of the submitting system, we use the downloading manuscript indicating the page and the line (page-line) to facilitate the work of the reviewers.
Looking forward hearing from you soon.
Sincerely,
Qiyu Luo and Yong Chen
Response to comments
- It is recommended not to use the active form of the verb in scientific articles. Please correct this issue in the abstract and whole manuscript.
We agree and we revise it.
- Please add keywords such as Agrobacterium, goosegrass, genetic transformation, etc. to broaden the visually of the article for various readers.
We agree, and we add the information of keywords as follows: “Genetic transformation; Agrobacterium; Herbicide stress; Paraquat-resistance; Goosegrass”
- Abstract is focused on results and did not provide a good overview of the goal and final conclusion. It is highly recommended to revise this important part.
We agree and revise it as follows: “Then, using this transformation method, we overexpressed the paraquat-resistant EiKCS gene into a paraquat-susceptible goosegrass biotype MZ04 and confirmed the stable inheritance of paraquat-resistance in the transgenic goosegrass lines. It may provides a potential mechanism for the evolution of paraquat-resistant goosegrass and a promising gene for the manipulation of paraquat-resistance plants.”
- In the title of the manuscript paraquat is given but in the introduction less details is provided. Please revise the introduction based on this issue.
We agree and add more information of paraquat as follows: “Paraquat (1,1’-dimethyl-4,4’-bipyridinium dichloride) is one of the most widely used broad-spectrum, quick-acting, and nonselective herbicide[18]. Although a mutation of DTX6 was confirmed in Arabidopsis to enhance paraquat resistance, the genetic loci of weeds for paraquat resistance still lack the identification [19,20] .”
- Introduction is too short and it is highly recommended to add more details and background related to the done research.
We agree and add more information of details and background as follows: “However, it found that the structure of paraquat is similar to that of polyamines, which could share a common uptake system endogenous substrates due to their structural similarity [21–24]. Recently, three genes on polyamine uptake transportation was applied to enhance the tolerance of paraquat-resistance in rice [25]. Therefore, it's worth exploring the mechanism of polyamine genes regulating paraquat resistance.”
- Please provide a graphical overview of the vector construction and transformation system for goosegrass.
We agree and add a graphical overview of the vector construction and transformation system for goosegrass in the “4.3. Vector construction for overexpressing EiKCS” of Materials and Methods followed as Figure 7.
- Materials and methods section is too long and in some parts it looked like a result. It is recommended to revise this section and use appendix for extra information.
We agree and improve it. But the manuscript was initially written as a method paper. Because this text contained a more detailed description of the method to aid readers in its application and replication, thus, it aimed to offer a tool or way for investigating the mechanism of environmental stress in weeds. Or it may depend on whether the journal could move the materials and methods section to an appendix for extra information.

Reviewer 2 Report
The authors reported a method to efficiently induce calli from seeds and first constructed transgenic lines in goosegrass. The manuscript has some major issues and is missing some details that might cause confusion to the audience.
1. Lines 76-77. The authors described seeds have the highest callus induction, but it is really hard to see from the figure. Could you provide a statistical analysis? When reading Figure 1A, the stem looks like having higher efficiency than seeds.
2. Lines 80-82. The genotype MZ04 has an average induction rate above 75%, however, in the best media N6, the highest induction rate is 60.09%. It is confusing when the authors had below 60% induction rates for MZ04 in Figure 1D, but claimed that they are more than 75% induction rates in Figure 1C.
3. Line 83. The authors classified calli from type I to type III. What are your criteria for this classification? Is there specific terminology about calli types or did the authors define the types by yourself? If it is a term, please give some background information in the introduction, because not everyone knows callus types. If the authors defined the types on their own, please provide details in the methods.
4. Line 112, please spell out 2,4-D since it is the first time this chemical is mentioned in the manuscript.
5. Section 2.2, please provide the rationale for choosing those culture conditions (NaOH, cold treatment, ethanol, etc.) to induce calli or break seed dormancy.
6. Figure 5A, it is hard to tell wildtype died completely at 90 DAT as I can see some green leaves in the pots. Could the authors provide statistical analysis for this?
7. Figure 5B, the authors labeled control plants (I think they are controls as the red lines indicated (-) paraquat), but there are no such plants in the figure.
8. Figure 6A the last panel about endogenous EiKCS expression is confusing. It seems to me both transgenic T6 lines, susceptible and resistant plants all expressed the same levels of EiKCS gene, then how come they will have different resistant phenotypes? Please clarify.
9. Line 245 and Figure 6B, C, please provide the rationale for adding the treatment paraquat+spermidine. Why the expression patterns of OE-EiKCS are opposite in transcript and protein levels?
10. lines 260-264, and Figure 6D. I don't understand what the authors were trying to convey in this experiment. My understanding is that the authors wanted to show spermidine as polyamine reduced EiKCS so the plants are more susceptible to paraquat. I guess my confusion is why the authors think polyamine impacts EiKCS. What is the effect of spermidine alone on the plants? And I don't see a difference between with and without spermidine treatments in all the plants. It will be great if the authors can add statistical analysis.
11. I noticed the authors used a significantly lower concentration of paraquat (270 mg/L) in the experiment in Figure 6 compared to previous results in Figure 5. And the susceptible genotype is MZ04, whose paraquat tolerance is unknown, Figure 5C only showed JM died in low paraquat previously. Do MZ04 and JM have the same genetic background?
12. The text font and size are not consistent throughout the manuscript, please check and fix them.
Author Response
RESPONSE TO REVIEWERS
of “ Establishment of an Efficient Agrobacterium-Mediated Genetic Transformation System to Enhance the Tolerance of the Paraquat Stress in Engineering Goosegrass (Eleusine indica L.)”
by Qiyu Luo, Shu Chen, Hai Nian, Qibing Ma, Yuyao Ding, Qinwen Hao, Jiping Wei, Jinesh D. Patel, Joseph Scott McElroy, Yaoguang Liu , and Yong Chen,
21th of March of 2023
We would like to thank you for your thoughtful review of the manuscript. You raise important issues are very helpful for improving the manuscript. We agree with almost all your comments and we have revised our manuscript accordingly.
We respond below in detail to each of the reviewer’s comments. In addition, we include how we have revised things, or if we have slightly disagreed with something, we stated why. We are already crafting a revised version of the paper that states the methods and results of our work more clearly than before. Moreover, We are willing to finish the revised version of the manuscript including any further suggestions that the reviewers may have.
Please, find below your comments repeated in italics and our responses inserted after each comment. In the requirement of the submitting system, we use the downloading manuscript indicating the page and the line (page-line) to facilitate the work of the reviewers.
Looking forward to hearing from you soon.
Sincerely,
Qiyu Luo and Yong Chen
Response to comments
- Lines 76-77. The authors described seeds have the highest callus induction, but it is really hard to see from the figure. Could you provide a statistical analysis? When reading Figure 1A, the stem looks like having higher efficiency than seeds.
Yes, the stem looks like having higher efficiency, but the efficiency of callus induction meant the embryogenic calli of goosegrass. The calli from the stems with low quality were hard to use in subsequent infection experiments, and they were hard to grow on subculture media. Later, we explain the embryogenic calli of type I to â…¢ of goosegrass callus.
And we add the information in the introduction to a better understanding as follows: “In terms of callus types, they were defined and classified from type I to type III in switchgrass cultivars [17]. And we found these types were consistent with the callus types of goosegrass.”
- Lines 80-82. The genotype MZ04 has an average induction rate above 75%, however, in the best media N6, the highest induction rate is 60.09%. It is confusing when the authors had below 60% induction rates for MZ04 in Figure 1D, but claimed that they are more than 75% induction rates in Figure 1C.
Yes, it was confusing because these tests were conducted simultaneously. Figure 1B, 1C, and 1D represents complete, simultaneous, independent experiments with different conditions. The seeds (S-HN goosegrass) were used in the best media N6 (Figure 1D, below 60.09%) when we didn’t know the seeds of MZ04 had a higher induction rate(78.86%). The frequency in Figure 1C meant most induction rates of the MZ04 seeds were distributed above 75% of all batches with an induction rate of 78.86%.
- Line 83. The authors classified calli from type I to type III. What are your criteria for this classification? Is there specific terminology about calli types or did the authors define the types by yourself? If it is a term, please give some background information in the introduction, because not everyone knows callus types. If the authors defined the types on their own, please provide details in the methods.
We agree and add the background information about callus types as follows: “In terms of callus types, they were defined and classified from type I to type III in switchgrass cultivars [17]. And we found these types were consistent with the callus types of goosegrass.”
- Line 112, please spell out 2,4-D since it is the first time this chemical is mentioned in the manuscript.
We agree and revise it as follow: “2,4-dichlorophenoxyacetic acid”.
- Section 2.2, please provide the rationale for choosing those culture conditions (NaOH, cold treatment, ethanol, etc.) to induce calli or break seed dormancy.
We agree and add some previous reports as follows: “ To break the seed dormancy, the goosegrass seeds were respectively treated by ddH2O (S-HN), 30% NaOH (S-HN and R-NX), -20℃ (S-HN), 4℃ (S-HN) and 75% ethanol (S-HN), according to the previous reports [39, 40]. ”
- Figure 5A, it is hard to tell wildtype died completely at 90 DAT as I can see some green leaves in the pots. Could the authors provide statistical analysis for this?
Yes, the wild-type goosegrass died two days after paraquat treatment, but the new plants grew up from other seeds in the soil after a long time at 90 DAT (three months). To continuously observe the resistant phenotypes of the transgenic goosegrass (T2 generation grass), we maintained all the original states after treatments for propagation. Although other new wild-type goosegrass showed some green leaves, we still left them alone.
We agree and add statistical analysis for the progeny of goosegrass with stable genetics (Figure 6). And we provided a graphical overview of goosegrass propagation.
- Figure 5B, the authors labeled control plants (I think they are controls as the red lines indicated (-) paraquat), but there are no such plants in the figure.
Yes, they are controls. We revised it by indicating the control (-) with black lines and the paraquat (+) with red lines. And we also slightly moved lines into the accurate sites in Figure 5B.
- Figure 6A the last panel about endogenous EiKCS expression is confusing. It seems to me both transgenic T6 lines, susceptible and resistant plants, susceptible and resistant plants all expressed the same levels of EiKCS gene, then how come they will have different resistant phenotypes? Please clarify.
We agree. It was difficult to distinguish the copy numbers of genes among the transgenic lines, and susceptible and resistant plants by merely using an ordinary PCR method in Figure 6A. Because the number of PCR cycles was 35X, which had reached the saturation amplification of the gene in all materials. The PCR method just was used to detect the specific fragment for screening transgenic EiKCS goosegrass. Furthermore, Real-time quantitative RT-PCR was used to detect the over-expressing levels of the EiKCS gene.
- Line 245 and Figure 6B, C, please provide the rationale for adding the treatment paraquat+spermidine. Why the expression patterns of OE-EiKCS are opposite in transcript and protein levels?
We agree and rewrite it. And we provide the information in the Introduction as follows: “However, it found that the structure of paraquat is similar to that of polyamines, which could share a common uptake system endogenous substrates due to their structural similarity [21–24]. Recently, three genes on polyamine uptake transportation was applied to enhance the tolerance of paraquat-resistance in rice [25]. Therefore, it's worth exploring the mechanism of polyamine genes regulating paraquat resistance” and “And we found the treatments of exogenous spermidine appeared to protect the paraquat-susceptible goosegrass in response to salt stress [28] ”.
Yes, it’s a thoughtful question. We also thought the expression patterns of OE-EiKCS were opposite in transcript and protein levels. But the current information in this study could indicate that the expression pattern of EiKCS had feedback regulation to the expression pattern of EiKCS in transgenic goosegrass.
- lines 260-264, and Figure 6D. I don't understand what the authors were trying to convey in this experiment. My understanding is that the authors wanted to show spermidine as polyamine reduced EiKCS so the plants are more susceptible to paraquat. I guess my confusion is why the authors think polyamine impacts EiKCS. What is the effect of spermidine alone on the plants? And I don't see a difference between with and without spermidine treatments in all the plants. It will be great if the authors can add statistical analysis.
We did not undergo the damage treatments to propagate the seeds as the materials since transgenic goosegrass had survived under treatments. The harvested seeds will be used as materials for the next generation to undergo resistance tests. The specific mechanisms of paraquat resistance will be shown in our future research.
We agree and takes much time to repeat the experiments in Figure 6D to add statistical analysis. And add the information in both of the Introduction “And we found the treatments of exogenous spermidine appeared to protect the paraquat-susceptible goosegrass in response to salt stress [28]”and the Results “The fresh weights of goosegrass were statistically analyzed at 48 h after these treatments (Figure 6E). The results showed that paraquat-susceptible goosegrasses were more significantly inhibited by paraquat, while the transgenic grass improved its tolerance to paraquat”.
- I noticed the authors used a significantly lower concentration of paraquat (270 mg/L) in the experiment in Figure 6 compared to previous results in Figure 5. And the susceptible genotype is MZ04, whose paraquat tolerance is unknown, Figure 5C only showed JM died in low paraquat previously. Do MZ04 and JM have the same genetic background?
We agree and revise it. We addressed the paraquat tolerance in “4.1. Plant materials” of Materials and Methods as follows: Among 16 biotypes of goosegrass, the MZ04 biotype was the most susceptible to paraquat (GR50 24.29 g a.i.ha-1), while the QY05 biotype (GR50 314.43 g a.i.ha-1) was the most resistant. Due to only the results of JM research in Figure 5C, we added the results of MZ04 research in Figure 6 as the control. The MZ04 and transgenic goosegrass (OE-EiKCS) have the same genetic background.
- The text font and size are not consistent throughout the manuscript, please check and fix them.
We agree and check them. We downloaded the latest version of the manuscript for revision from the submitting system and fixed the text font and size.

Round 2
Reviewer 1 Report
Thanks for your response.
Reviewer 2 Report
Thanks for the authors' responses. My comments were addressed and the revised manuscript is improved.